# Povidone-Iodine as a Pre-Procedural Mouthwash to Reduce the Salivary Viral Load of SARS-CoV-2: A Systematic Review of Randomized Controlled Trials

**DOI:** 10.3390/ijerph19052877

**Published:** 2022-03-01

**Authors:** Alvaro Garcia-Sanchez, Juan-Francisco Peña-Cardelles, Esther Ordonez-Fernandez, María Montero-Alonso, Naresh Kewalramani, Angel-Orión Salgado-Peralvo, Dániel Végh, Angélica Gargano, Gabriela Parra, Lourdes-Isabela Guerra-Guajardo, Wataru Kozuma

**Affiliations:** 1Department of Oral Health and Diagnostic Sciences, School of Dental Medicine, University of Connecticut Health, Farmington, CT 06030, USA; 2Department of Health Sciences, Rey Juan Carlos University, 28040 Madrid, Spain; juanfranciscopenacardelles@gmail.com; 3Oral and Maxillofacial Surgery Department, School of Dental Medicine, University of Connecticut Health, Farmington, CT 06030, USA; 4Department of Prosthodontics, School of Dental Medicine, University of Connecticut Health, Farmington, CT 06030, USA; w.kouzuma@gmail.com; 5Division of General Dentistry, Department of Craniofacial Sciences, School of Dental Medicine, University of Connecticut Health, Farmington, CT 06030, USA; orfe.esther22@gmail.com (E.O.-F.); angelicatmunoz93@gmail.com (A.G.); drparrabarillas@gmail.com (G.P.); lourdes.iguerrag@gmail.com (L.-I.G.-G.); 6Pediatric Service, Hospital Álvaro Cunqueiro, 36213 Vigo, Spain; maria.montero.alonso@gmail.com; 7Department of Nursery and Stomatology, Rey Juan Carlos University, 28922 Madrid, Spain; k93.naresh@gmail.com; 8Department of Stomatology, Faculty of Dentistry, University of Seville, 41009 Seville, Spain; 9Department of Prosthodontics, Semmelweis University, 1085 Budapest, Hungary; vegh.daniel.official@gmail.com; 10Division of Oral Surgery and Orthodontics, Department of Dentistry and Oral Health, Medical University of Graz, 8010 Graz, Austria; 11Department of Regenerative and Reconstructive Dental Medicine, Tokyo Medical and Dental University, 1-5-45 Yushima, Bunkyo-ku, Tokyo 113-8549, Japan

**Keywords:** COVID-19, SARS-CoV-2, mouthwashes, mouthrinse, aerosols, chlorhexidine, povidone-iodine, cetylpiridinium chloride, hydrogen peroxide, colony-forming units

## Abstract

The use of pre-procedural rinses has been investigated to reduce the number of viral particles and bacteria in aerosols, potentially decreasing the risk of cross-infection from severe acute respiratory syndrome coronavirus 2 (SARS-CoV-2) during medical and dental procedures. This review aims to confirm whether there is evidence in the literature describing a reduction in salivary load of SARS-CoV-2 when povidone-iodine (PVP-I) is used as a pre-intervention mouthwash. An search of the MEDLINE, Embase, SCOPUS, and the Cochrane library databases was conducted. The criteria used followed the PRISMA^®^ Statement guidelines. Randomized controlled trials investigating the reduction of salivary load of SARS-CoV-2 using PVP-I were included. Ultimately, four articles were included that met the established criteria. According to the current evidence, PVP-I is effective against SARS-CoV-2 in saliva and could be implemented as a rinse before interventions to decrease the risk of cross-infection in healthcare settings.

## 1. Introduction

Aerosols are defined as particles of liquid or solid in gas and are ≤5 μm in diameter [1]. Due to their small size, they are inspirable and can remain suspended in the air for hours [2]. These particles are generated by aerosol-generating procedures (AGPs) in medical settings, including airway suctioning, bronchoscopies, and high-flow oxygen therapy, among many others [3]. Subsequently, aerosols in dental offices are generated by the frequent use of high-speed handpieces, ultrasonic devices, and 3-in-1 air-water syringes. For this reason, dentists are one of the collectives that have the highest risk of infection of COVID-19 due to the close proximity with the patients’ oral cavities and the numerous AGPs performed routinely [4]. Saliva and blood are main components for viral and bacterial spread; therefore, procedures that generate aerosols should be minimized. However, dental clinicians have a particularly limited range of options regarding treatments and armamentarium that do not generate aerosols [4].

The primary mode of transmission of severe acute respiratory syndrome coronavirus 2 (SARS-CoV-2) is through aerosols and respiratory droplets generated during daily activities, such as speaking or coughing [5]. Several factors, including the immune response of the host, the pathogenicity of the virus, and the amount of infected particles, determine the susceptibility of being infected via an aerosol [6,7,8]. Furthermore, it has been demonstrated that COVID-19-positive patients present high viral loads in saliva [9,10]; therefore, healthcare professionals performing AGPs have a greater risk of becoming infected with SARS-CoV-2 [11]. 

For this reason, there have been several investigations attempting to mitigate the negative effects of aerosols during AGPs. Pre-procedural rinses have been explored to reduce the salivary load of different microorganisms and the colony-forming units (CFUs) in aerosols, which could potentially decrease the risk of cross-infection during medical and dental procedures [9,12,13,14,15]. One of the most predominant mouthwash solutions studied is povidone-iodine (PVP-I). This molecule is an iodophor globally used due to its broad-spectrum antiseptic properties with a low number of contraindications, including allergy to iodine, thyroid disease, and pregnancy [16]. Several in vitro studies and, more recently, randomized controlled trials (RCTs) using PVP-I as pre-procedural mouthwash have published their results.

Therefore, this study aimed to investigate the effectiveness of PVP-I used as a mouthwash to decrease the salivary viral load of SARS-CoV-2.

## 2. Materials and Methods

### 2.1. Protocol

This review was performed according to the Preferred Reporting Items for Systematic Reviews and Meta-Analyses (PRISMA^®^) Statement [17,18]. The protocol was registered at the International Prospective Register of Systematic Reviews (PROSPERO) under the registration number CRD42022303756.

### 2.2. Focused Question

A PICO (P, population/patient/problem; I, intervention; C, comparison; O, outcome) question was formulated based on the PRISMA^®^ guidelines:

“In patients diagnosed with COVID-19 (P), does the use of PVP-I mouthwashes (I) compared to not prescribing them (C) reduce the viral load present in saliva (O)?”

### 2.3. Eligibility Criteria

Prior to the search, inclusion and exclusion criteria were defined:

#### 2.3.1. Inclusion Criteria

Included studies were (a) RCTs; (b) studies in which the participants had a reverse-transcription polymerase chain reaction (RT-PCR) examination positive for SARS-CoV-2; (c) studies that used PVP-I as a form of intervention; and (d) studies published in English. 

#### 2.3.2. Exclusion Criteria

Excluded studies were the following: (a) animal studies; (b) experimental laboratory studies; (c) studies whose study base focused on other areas besides the oral cavity and/or oropharynx; (d) studies that did not evaluate the reduction of viral load in saliva; (e) non-RCTs; (f) systematic reviews and meta-analyses; (g) literature review studies; (h) case reports; (i) letters to the editor; (j) abstracts or conference papers; (k) comments; and (l) unpublished articles.

### 2.4. Information Sources and Search Strategy

The search was conducted in four different electronic databases: MEDLINE (via PubMed), Embase, SCOPUS, and the Cochrane library database.

The search strategy was carried out by two researchers independently (A.G.-S. and A.-O.S.-P.). The search was not time-restricted and was updated to January 2022. MeSH (Medical Subjects Headings) terms, keywords, and other free terms were used with Boolean operators (OR, AND) to merge searches: (‘povidone’ OR ‘povidone-iodine’ OR ‘polyvidone iodine’ OR iodopovidone’ OR ‘PVP-I’ OR ‘iodine’) AND (‘COVID-19’ OR ‘SARS-CoV-2’ OR ‘SARS’). These keywords were implemented in all databases according to the syntax rules of each database.

### 2.5. Study Records

The results were independently compared by two authors (A.G.-S. and A.-O.S.-P.) to guarantee completeness and removal of duplicates. Next, the title and abstract of the remaining articles were screened individually. Ultimately, full-text papers to be included in this study were selected following the criteria previously described. Disagreements over eligible articles were resolved by including a third author (J.-F.P.-C.) to reach a consensus. 

### 2.6. Risk of Bias Assessment

The methodology of eligible studies was evaluated following the Joanna Briggs Institute (JBI) Critical Appraisal Tool [19] by two independent authors (A.G.-S. and A.-O.S.-P.). The studies were categorized as low-quality (0–7 domains) or high-quality assessment (8–13 domains). A third author (J.-F.P.-C) was included to resolve any disagreements between the two authors. 

## 3. Results

### 3.1. Study Selection

The search strategy resulted in 630 articles. There were 218 duplicates; therefore, 412 remained. Then, two authors (A.G.-S. and A.-O.S.-P.) independently examined the titles and abstracts and excluded 375 articles that were beyond the scope of this study. Therefore, we obtained 37 possible references. After reading the full text of those 37 papers, 33 were excluded because they investigated areas other than oral cavity saliva and/or oropharyngeal saliva (*n* = 6) or were experimental laboratory studies (*n* = 9), systematic reviews (*n* = 1), literature reviews (*n* = 2), letters to the editor (*n* = 12), and commentaries (*n* = 3). Therefore, four studies were included in our systematic review (Figure 1). 

### 3.2. Study Characteristics

All the studies included were RCTs published in 2020 and 2021. There are some discrepancies in the sample size of the different articles (ranging from 36 to 84). Due to the low number of studies available, it was decided that there would not be exclusion criteria set for a minimum of participants. The total number of patients included within the studies was 221. All patients recruited had a positive RT-PCR examination for SARS-CoV-2.

In these studies, rinsing times ranged between 30 s and 1 min. In the placebo groups, distilled water [20,21,22] and saline [23] were used. In the test group, several concentrations of PVP-I were used: 0.50% [21,23], 1% [22], and 2% [20]. All saliva samples were evaluated with RT-PCR. Baseline samples were collected immediately before the interventions. The number of saliva samples after interventions varies among the studies. One study collected one sample of saliva after intervention [22], one collected two samples [23], and two collected three samples [20,21]. 

A summary of the findings of the included articles is described in Table 1.

The central findings of the resulting articles are described as follows:

Chaudhary et al. [23] (2021) evaluated the effect of 0.50% PVP-I, 1% hydrogen peroxide (HP), 0.12% chlorhexidine (CHX), and normal saline. Forty patients were randomly allocated into groups, and saliva samples were collected and tested with RT-PCR. In the PVP-I group, there was a median reduction of 61% and 97% at 15 and 45 min, respectively.

Seneviratne et al. [21] (2020) recruited 36 patients that were randomly allocated to four different groups: distilled water (control), 0.50% PVP-I, 0.20% CHX, and 0.075% cetylpyridinium chloride (CPC). Samples were collected before rinsing, 5 min, and 3 and 6 h post-rinse. Cycle threshold (Ct) changes were estimated at each time-point value. The PVP-I group exhibited greater changes in Ct values after 5 min and 3 h; however, there was a significant increase in the virucidal activity only at 6 h when compared to distilled water.

Elzein et al. [22] (2021) performed a triple-blinded RCT evaluating 1% PVP-I, 0.20% CHX, and distilled water (control) in 61 patients. Saliva samples were collected at baseline and 5 min post-rinse. The delta Ct values (4.72 ± 0.89) indicated a statistically significant reduction in the salivary viral load after using 1% PVP-I for 30 s compared to distilled water (0.519 ± 0.519). 

Ferrer et al. [20] (2021) evaluated the differences in viral load of SARS-CoV-2 in 80 participants using 2% PVP-I, 1% HP, 0.07% CPC, 0.12% CHX, and distilled water (control). Samples were collected at baseline, 30, 60, and 120 min post-rinse. There was not a statistically significant reduction in salivary load when 2% PVP-I was used compared with the control group.

### 3.3. Risk Bias Assessment

Using the JBI Critical Appraisal Tool for RCTs [19], we determined that none of the articles presented a low-quality assessment (0–7 domains), and all of the articles included [20,21,22,23] had a high-quality assessment (8–13 domains). Table 2 shows a detailed description of the studies included.

## 4. Discussion

The surface of SARS-CoV-2 presents a spike protein (S) involved in the receptor recognition and cell membrane fusion process. The S protein mediates cell entry when it contacts the angiotensin-converting enzyme 2 (ACE2) receptors [24], and oral mucosa and salivary gland epithelium present a great amount of these receptors [25,26,27]. In a study by Huang et al. [28], RNA molecules of SARS-CoV-2 were consistently found in ACE2-expressing ducts of salivary glands and in epithelial cells of the oral mucosa. They also proposed that the virus replicating in infected glands and the shedding of the infected oral mucosa are the sources of SARS-CoV-2 in saliva [28].

Patients undergoing medical and dental procedures were thoroughly screened for COVID-19 signs and symptoms as a means to prevent the risk of infection of healthcare providers. However, SARS-CoV-2 mRNA was detected in saliva of asymptomatic/pre-symptomatic patients [28]. For that reason, it might be beneficial to use mouthwashes, such as PVP-I, to decrease the risk of cross-infection in healthcare settings where AGPs are performed in both confirmed COVID-19 and asymptomatic patients.

PVP-I efficacy in reducing the salivary viral load was compared to other solutions in the articles included in this review. Chaudhary et al. [23] found that reduction in viral load at 15 and 45 min did not differ among 1% HP, 0.12% CHX, and 0.50% PVP-I. Similarly, Elzein et al. [22] did not find any significant difference in the reduction of salivary load between 0.20% CHX and 1% PVP-I, and both were significantly effective compared to distilled water. In the RCT by Seneviratne et al. [21], salivary Ct values within all groups at the different time periods did not demonstrate any significant differences. Nonetheless, compared to distilled water, CPC was significantly more effective at 5 min and 6 h, while PVP-I was only significantly more effective at the 6-h mark. Ferrer et al. [20] found no statistically significant changes in salivary load of SARS-CoV-2 in any of the mouthwashes evaluated. However, comparing the loads at baseline and after intervention, PVP-I and CPC groups showed mean reductions of 30%, with the highest activity 2 h after intervention. None of the articles included showed any complications after oral PVP-I use at different concentrations.

Various in vitro studies have also assessed the virucidal activity of PVP-I. Many studies used the logarithmic reductions scales of viral load in their results. As a reference, a 3log_10_ reduction equals a 99.90% reduction, and a 4log_10_ reduction equals a 99.99% reduction in viral load. Xu et al. [29] found a virucidal activity of >3log_10_ with a contact time of 30 min. Hassandarvish et al. [30] evaluated the reduction in salivary viral load using PVP-I at concentrations of 1% and 0.50%, which resulted in virucidal activities of >5log_10_ (> 99.99%) at 15 and 30 s, respectively. Similar studies found virucidal activities of >4log_10_ at 15 [31], 30 [32], and 60 s [33,34].

When evaluating PVP-I as a nasal rinse, Frank et al. [35] showed a complete inactivation of the virus using 0.50% PVP-I with a contact time of 15 s. Furthermore, a RCT study by Guenezan et al. [14] evaluated the reductions of viral titers in the nasopharynx using a 1% PVP-I rinse followed by 1% PVP-I nasal spray and an application of a 10% PVP-I balm over nasal mucosa during 7 days. The mean reductions in salivary load were 75% at day 1 compared with a reduction of 32% in the placebo group. However, there was no difference in the reduction of viral load over 7 days. 

The results of this systematic review show that PVP-I is a potentially effective pre-procedural mouthwash to decrease the salivary viral load of symptomatic and asymptomatic COVID-19-positive patients. The prevention of the asymptomatic transmission of SARS-CoV-2 is still one of the biggest challenges [36], and the implementation of protocols to reduce the salivary load of SARS-CoV-2 before AGPs could play a significant role in decreasing the risk of cross-infection in healthcare settings.

### 4.1. Strengths and Limitations

This systematic review presents several strengths, including an unrestricted search in the literature, the search protocol, data retrieval, and risk assessment analysis performed in duplicate.

However, COVID-19 is a disease that is continuously being investigated, and multiple RCTs are evaluating the use of different mouthwashes in progress at this moment. In addition, this systematic review only included four RCTs; therefore, our results must be interpreted with caution, and further investigations must be carried out soon.

### 4.2. Recommendations for Further Research

This study systematically reviewed the first RCTs investigating PVP-I as a pre-procedural rinse. Further in vitro studies evaluating potential new mouthwash solutions and additional RCTs are needed to demonstrate the safety and efficacy of different mouthwashes.

## 5. Conclusions

Within the limitations of this study, PVP-I presents a significant virucidal activity against SARS-CoV-2 in saliva with concentrations ranging from 0.5% to 1%. On the other hand, concentrations of 2% did not show statistically significant changes in salivary load in one of the included studies. In clinical practice, a 30- or 60-s pre-procedural rinse of 0.50/1% PVP-I could be beneficial to reduce the risk of cross-infection in healthcare settings performing AGPs in diagnosed, suspected, or asymptomatic COVID-19-positive patients. However, these results should be taken with caution, as this review included a low number of studies, and additional RCTs are essential to confirm the validity of these findings.

## Figures and Tables

**Figure 1 ijerph-19-02877-f001:**
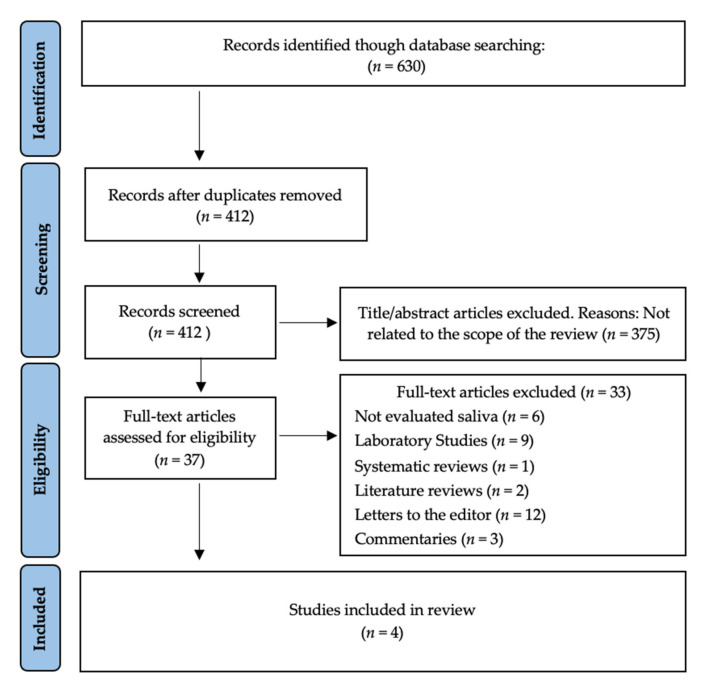
PRISMA^®^ flow diagram of the search processes and results.

**Table 1 ijerph-19-02877-t001:** Results of the included RCTs.

Author/Year	Sample Size	Time of Testing	Intervention/Duration of Rinses		Conclusions
			Control Group	Test Group(s)	
Chaudhary et al. [23] (2021)	40	Saliva samples for RT-PCR ^1^ were collected at 15 and 45 min ^2^ post-rinse.	Placebo (normal saline), 1%/ 60 s ^3^	1% HP ^4^, 0.12% CHX ^5^, 0.50% PVP-I ^6^.Rinsed with 15 mL ^7^/for 60 s.	All 4 mouthwashes reduced the salivary load by 61% through 89% at 15 min and by 70% through 97% at 45 min.
Elzein et al. [22] (2021)	61	Saliva was evaluated with RT-PCR at baseline and 5 min after rinsing.	Placebo (distilled water)/ 30 s.	1% PVP-I and 0.20% CHX/30 s.	The Ct ^8^ of the intervention groups (CHX 0.20% and 1% PVP-I) was significantly different compared to the control group.
Ferrer et al. [20] (2021)	84	RT-PCR at baseline, 30, 60, and 120 min after mouth rinse	Placebo (distilled water)/ 1 min.	2% PVP-I, 1% HP, 0.07% CPC ^9^, 0.12% CHX/1 min.	None of the mouthwashes evaluated presented a statistically significant change in the salivary viral load.
Seneviratne et al. [21] (2020)	36	Saliva samples for RT-PCR at baseline (pre-rinse), 5 min, and 3 and 6 h ^10^ post-rinsing.	Placebo (water)/ 30 s.	0.5% PVP-I, 0.20% CHX, 0.075% CPC/30 s.	There were no differences in the reduction of salivary load in all intervention groups. When compared with the control group, PVP-I and CPC showed a significant decrease at 6 h. CPC also showed a significant reduction at 5 min.

^1^ RT-PCR, reverse-transcription polymerase chain reaction; ^2^ min, minutes; ^3^ s, seconds; ^4^ HP, hydrogen Peroxide; ^5^ CHX, chlorhexidine; ^6^ PVP-I, povidone-iodine; ^7^ mL, milliliters; ^8^ Ct, cycle threshold; ^9^ CPC, cetylpyridinium chloride; ^10^ h, hour(s).

**Table 2 ijerph-19-02877-t002:** JBI Critical Appraisal Tool [19] for RCTs. Reprinted with permission from JBI. Copyright 2020.

Critical Appraisal Questions	Chaudhary et al. [23] (2021)	Seneviratne et al. [21] (2020)	Elzein et al. [22] (2021)	Ferrer et al. [20] (2021)
Was true randomization used for the assignment of participants to treatment groups?				
2.Was allocation to treatment groups concealed				
3.Were treatment groups similar at the baseline?		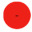		
4.Were participants blind to treatment assignment?		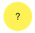		
5.Were those delivering treatment blind to treatment assignment?		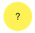		
6.Were outcome assessors blind to treatment assignment?		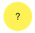		
7.Were treatment groups treated identically other than the intervention of interest?				
8.Was follow up complete and if not, were differences between groups in terms of their follow-up adequately described and analyzed?				
9.Were participants analyzed in the groups to which they were randomized?	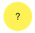	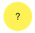	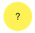	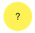
10.Were outcomes measured in the same way for treatment groups?				
11.Were outcomes measured in a reliable way?				
12.Was appropriate statistical analysis used?				
13.Was the trial design appropriate and any deviations from the standard RCT design accounted for in the conduct and analysis of the trial?				


 = yes, 
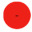
 = no, 
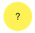
 = uncertain.

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
