# Peer review of "Povidone-Iodine as a Pre-Procedural Mouthwash to Reduce the Salivary Viral Load of SARS-CoV-2: A Systematic Review of Randomized Controlled Trials"

_ijerph, 2022, doi:10.3390/ijerph19052877_

Round 1

Reviewer 1 Report

I would like to congratulate the authors on the contemporary idea. The article provides an in-depth analysis of the literature related to clinical trials on the use of PVP-I as a mouthwash for reducing the salivary load of SARS-CoV-2. The review is concise and performed according to the PRISMA® guidelines.

I have only one remark: In table 1, where the RCT of Chaudhary et al. is discussed, under “Control group” is written: Placebo (normal saline), 1%/60 s. I would like to ask what the meaning of “1%” is?

Author Response

Dear Reviewer,

Thank you for taking the time to review our manuscript.

  1. 1. I have only one remark: In table 1, where the RCT of Chaudhary et al. is discussed, under “Control group” is written:Placebo (normal saline), 1%/60 s. I would like to ask what the meaning of “1%” is?

Response: Thank you for pointing this out. It was a typographical error. Normal saline by definition is 0.9% w/v of NaCl, but we decided to remove the percentage to avoid redundancy and confusion. We have seen that “normal” saline is accepted without the percentage as it already implies the percentage in its name.

Thank you again.

Reviewer 2 Report

Dear Authors,

the topic is very actual, the research is well designed and carried out. 

Introduction contains enough background informations, but I suggest to extend the paragraph regarding AGPs, considering dental procedures that, in general, often produces a lot of aerosol.

Materials and methods are clearly described and your systematic review is carried out following the protocols.

Results and discussion are presented in a proper way.

Conclusions could be improved and extended, please include some clinical considerations.

Author Response

Dear Reviewer,

Thank you for reviewing our manuscript and providing valuable feedback.

  1. 1. Introduction contains enough background information, but I suggest to extend the paragraph regarding AGPs, considering dental procedures that, in general, often produces a lot of aerosol.

Response: Thank you for your suggestion. We agree, therefore, we have extended the paragraph accordingly. We have added information regarding the dentists’ highest risk of exposure due to AGPs, the role of blood and saliva in aerosols, and the limited range of options that dental practitioners have to reduce AGPs overall. You can find this on page 2, lines 54-60

  1. Conclusions could be improved and extended, please include some clinical considerations.

Response: Thank you for your comment. We agree with this. We have extended the conclusion, adding further information regarding the successful concentrations and contact times of povidone-iodine as well as mentioning the limited efficacy of the 2% concentration in one of the studies included. We added, as recommended, an extended clinical recommendation for a pre-procedural rinse with this solution before performing AGPs. Finally, we included a recommendation to interpret our results with caution due to the low number of RCTs available.

Thank you for helping us improve the quality of the manuscript.

Reviewer 3 Report

This review is very nicely written. I have only two suggestions for the authors:

1) Within the limitation section add the information that the research was based only on 4 RCTs.

2) Conclusions should be corrected. The study by Ferrer et. al does not confirm that "PVP-I presents a significant virucidal activity against SARS-CoV-2 in saliva". You should critically review the literature and summarize the other authors' observations.

Therefore, I recommend acceptance after minor revision.

Author Response

Dear Reviewer,

Thank you for your time reviewing the manuscript and for your insightful comments.

  1. Within the limitation section add the information that the research was based only on 4 RCTs.

Response: Thank you for your comment. We agree with this, and we have included that information accordingly in the limitations section. You can find it on Page 7, lines 243-245.

  1. Conclusions should be corrected. The study by Ferrer et. al does not confirm that "PVP-I presents a significant virucidal activity against SARS-CoV-2 in saliva". You should critically review the literature and summarize the other authors' observations.

Response: Thank you for pointing this out. We agree with your recommendation, therefore, we have modified the conclusion accordingly to implement these changes, clarifying in the sentence that it could be beneficial as further studies are needed.

Thank you for your time and feedback.